# An Enhanced LightGBM-Based Breast Cancer Detection Technique Using Mammography Images

**DOI:** 10.3390/diagnostics14020227

**Published:** 2024-01-22

**Authors:** Abdul Rahaman Wahab Sait, Ramprasad Nagaraj

**Affiliations:** 1Department of Documents and Archive, Center of Documents and Administrative Communication, King Faisal University, P.O. Box 400, Hofuf 31982, Al-Ahsa, Saudi Arabia; 2Department of Biochemistry, S S Hospital, S S Institute of Medical Sciences & Research Centre, Rajiv Gandhi University of Health Sciences, Davangere 577005, Karnataka, India; ramprasad7u@gmail.com

**Keywords:** breast cancer, deep learning, EfficientNet, ensemble learning, feature engineering

## Abstract

Breast cancer (BC) is the leading cause of mortality among women across the world. Earlier screening of BC can significantly reduce the mortality rate and assist the diagnostic process to increase the survival rate. Researchers employ deep learning (DL) techniques to detect BC using mammogram images. However, these techniques are resource-intensive, leading to implementation complexities in real-life environments. The performance of convolutional neural network (CNN) models depends on the quality of mammogram images. Thus, this study aimed to build a model to detect BC using a DL technique. Image preprocessing techniques were used to enhance image quality. The authors developed a CNN model using the EfficientNet B7 model’s weights to extract the image features. Multi-class classification of BC images was performed using the LightGBM model. The Optuna algorithm was used to fine-tune LightGBM for image classification. In addition, a quantization-aware training (QAT) strategy was followed to implement the proposed model in a resource-constrained environment. The authors generalized the proposed model using the CBIS-DDSM and CMMD datasets. Additionally, they combined these two datasets to ensure the model’s generalizability to diverse images. The experimental findings revealed that the suggested BC detection model produced a promising result. The proposed BC detection model obtained an accuracy of 99.4%, 99.9%, and 97.0%, and Kappa (K) values of 96.9%, 96.9%, and 94.1% in the CBIS-DDSM, CMMD, and combined datasets. The recommended model streamlined the BC detection process in order to achieve an exceptional outcome. It can be deployed in a real-life environment to support physicians in making effective decisions. Graph convolutional networks can be used to improve the performance of the proposed model.

## 1. Introduction

BC is a neoplastic condition characterized by the formation of a malignant tumor originating from the breast cells [1]. It is the most prevalent form of cancer observed in women, with a lesser incidence in men across the globe [1,2]. In 2020, the International Agency for Research on Cancer reported more than 2.26 million new cases and about 685,000 mortalities globally [2]. The incidence of BC has increased over time [3]. After leukemia, BC is the leading cause of mortality in Saudi Arabia [3]. Globally, more than 100 million mammograms are performed annually to detect breast cancer [4]. Each mammography necessitates a minimum of two evaluations by expert radiologists to detect abnormalities and provide a detailed analysis of the mammographic image [4,5]. These factors render BC screening increasingly expensive and resource-intensive.

Chemotherapy and radiation therapy can significantly affect an individual’s physical and mental health [6]. Radiation or surgery can affect the lymphatic system, resulting in lymphedema [7]. Advanced treatment may cause a financial burden to individuals. As a result, individuals may face challenges in obtaining critical medical care. Earlier detection of BC can reduce its financial burden and BC’s mortality rate [8]. In addition, healthcare centers can provide personalized diagnoses based on BC stages.

Healthcare centers employ various techniques to screen and diagnose BC. Sound waves are used in ultrasound scanning techniques to represent breast tissue visually [8]. However, the expertise of the operators plays a crucial role in effective BC detection. Magnetic resonance imaging employs magnetic fields and radio waves to identify tumors [8]. It is widely used to detect abnormalities in advanced stages of BC. In addition, biopsy procedures are used to extract breast tissue specimens for BC staging [8]. Nonetheless, the invasive nature of this procedure may affect patients’ physical and mental health. 

Mammography is a specialized form of medical imaging that uses low-dose X-ray to visualize the internal organs of the breasts [8,9,10]. It is common practice to utilize mammography as a screening method for women who do not exhibit any symptoms of BC, particularly in age groups where the likelihood of acquiring BC is higher [11]. Regular screenings have the potential to detect anomalies prior to the manifestation of symptoms. There have been an extensive number of studies suggesting that routine mammographic screening programs lead to a reduction in BC mortality rates [12,13,14]. Mammography screening is affordable and readily accessible [14]. Advancements in imaging technologies have facilitated the rapid development of artificial intelligence (AI) for breast imaging. Computer-aided detection identifies potentially abnormal lesions, such as masses and microcalcifications [15]. On the other hand, computer-aided diagnosis analyzes a lesion’s suspicious region and estimates the likelihood of its occurrence.

The sensitivity, specificity, and capability to manage the complexity of mammogram images present potential obstacles to conventional BC screening approaches [16]. Early-stage BC patterns may be hidden in complex mammograms. The level of complexity exceeds the capabilities of the current image processing methods, which could cause diagnostic delays or false negatives (FNs) [17]. Traditional BC detection techniques are not generalizable to the broader population. They require substantial computational resources and expert opinions are needed to make a decision. 

Researchers have recently employed DL models for detecting BC with optimal outcomes [17]. The detection and diagnosis of BC can differ among radiologists due to variances in human mammography interpretation. DL models can learn complicated hierarchical features, potentially enhancing breast cancer detection sensitivity and specificity [18]. This facilitates an earlier identification of BC conditions and improves the precision of diagnostic procedures. The number of highly specialized professionals can be minimized using DL models [18]. A DL model can be integrated with edge and mobile devices to remotely screen individuals [18]. In addition, it can identify subtle signs of abnormalities in mammography images. In the context of BC diagnosis, it is possible to employ a pretrained CNN model on a substantial dataset, such as ImageNet, and subsequently fine-tune it using a smaller dataset consisting of mammography images [19]. Transfer learning (TL) can improve generalization using the pretrained model’s weights to identify BC’s low- and mid-level features [19]. Physicians can employ DL-based models to detect BC using mammography images. The models can handle a massive volume of images, which supports healthcare centers in diagnosing large numbers of individuals [19]. The pretrained models can extract critical features with limited resources. In addition, fine-tuning and image preprocessing techniques can improve the feature extraction process [20]. 

By automating the identification of BC with DL models, healthcare providers may process multiple mammograms in a shorter period, substantially enhancing screening efficiency. A lightweight BC detection model can be implemented in edge and mobile devices. Healthcare centers can benefit from this model and render their services in remote areas. The proposed study aims to offer reliable and unbiased assessments in BC detection. Thus, the authors build a DL-based BC detection model with limited computational resources. This study’s contributions are as follows: i.A feature engineering technique using QAT strategy to generate critical features of BC.

The authors built a CNN model with the EfficientNet B7 model’s weights. QAT strategy was used to improve the performance of the EfficientNet B7 model in a resource-constrained environment.

ii.Development of a fine-tuned LightGBM-based BC detection model.

The proposed feature engineering presents valuable features and passes them to the LightGBM model in order to detect BC from mammography images. In addition, the authors fine-tuned the LightGBM model using the Optuna algorithm.

The remaining parts of this study are organized as follows: Section 2 offers the research methodology for BC detection using mammography images. The experimental outcomes are outlined in Section 3. Section 4 discusses the significance of the proposed study in detecting BC. Lastly, Section 5 presents the features and limitations of the proposed research. 

## 2. Materials and Methods

The authors built a DL-based BC detection model using effective image preprocessing, augmentation, and feature extraction techniques. Image preprocessing techniques were used to improve the quality of mammography images. In addition, image augmentation techniques were employed to generate additional training samples and improve prediction accuracy. DL approaches offer the integration of numerous models in order to provide a final prediction. Improving the efficiency of CNN models can address complex problems, improve predictive accuracy, and provide deeper insights into the relationships between structured and unstructured data in various domains, including healthcare, finance, manufacturing, and more. Additionally, overall performance of the BC detection model was improved by utilizing a number of distinct models. Using critical features of BC images, the DL approach can learn minor details to predict BC.

The EfficientNet B7 model can learn complex patterns from its training data [21]. Compared to traditional CNN models, EfficientNet B7 offers an impressive performance with few parameters. It captures intricate image patterns and representations with top-tier performance on benchmark datasets. It offers remarkable efficacy across multiple image classification tasks, rendering it a comprehensive model suitable for medical imaging classification. In addition, in order to achieve a high level of precision, a compound scaling technique can be used to scale the depth and width of the model [22]. However, training the EfficientNet B7 model can be significantly resource-intensive and time-consuming. EfficientNet B7 may not be able to generalize effectively to completely new or unknown data. In a real-world setting, this constraint may affect model performance. Improving the EfficienNet B7 model’s performance can support the proposed model to generalize on real-time data. The LightGBM model encompasses the ensemble tree approach to handle the massive datasets [23]. 

The authors were motivated to construct a feature engineering model using EfficientNet B7 and employ the LightGBM model to predict BC using its features. Figure 1 presents the proposed BC detection model. 

### 2.1. Image Acquisition

In order to train the proposed BC detection model, the authors utilized the CBIS-DDSM and CMMD datasets, which are available in repositories [24,25]. The CBIS-DDSM dataset contains enhanced images of the DDSM dataset [26,27]. The dataset providers addressed the limitations of the DDSM dataset. They removed low-quality images and reannotated the images. They converted the quality of the primary images to 16-bit grayscale TIFF images. Spurious ground-labeled images were removed from the dataset. The interoperability and reusability of the dataset were improved by converting analog images into digitized images.

The CMMD dataset includes two datasets. CMMD1 encompasses mammography and clinical data, whereas CMMD2 covers malignant BC images. The dataset owner obtained the images from Sun-Yat-Sen University Cancer Center in Guangzhou and Nanhai Affiliated Hospital of Southern Medical University, Guangdong, China. The data were collected between July 2012 and July 2016. To ensure the efficiency of the proposed BC detection model and to use a unique dataset, the authors combined the CBIS-DDSM and CMMD datasets. They included benign and malignant images from the MIAS [28] and BCDR-D [29] datasets. Table 1 highlights the details of the datasets. Figure 2a,b show sample images from the CBIS-DDSM and CMMD datasets.

### 2.2. Image Preprocessing

Noises and artifacts can reduce the performance of BC detection models in identifying the abnormalities. Artifacts include grid lines, scratches, and distortions that can interfere with BC identification. These may mimic features related to lesions and tumors that may influence the BC detection model to generate false positives (FPs) and FNs. Image preprocessing techniques can improve the performance of the CNN models [20]. For instance, the EfficientNet model requires an image size of 224 × 224. In addition, image normalization helps pretrained models to extract key features of BC images. Thus, image preprocessing is required to improve EfficientNet B7-based feature extraction. The authors employed multiple image preprocessing techniques to overcome the shortcomings of BC images. Contrast-limited adaptive histogram equalization (CLAHE) was employed to enhance the local contrast of an image by preventing noise amplification. It applies histogram equalization to the smaller regions of BC images. Equation (1) presents the mathematical form of the image resizing process with the CLAHE method.
(1)$$I_{i}=Resize\_CLAHE\left(I_{i},c,t\right),\;i=1,\;2,\;\ldots \;n$$
where $I_{i}$ is the BC image, c is the clip limit, and t is the size of the tile grid.

### 2.3. Image Augmentation

In order to improve the training environment, the authors employed generative adversarial networks (GANs) [21,30]. They followed previous studies [21,30] to generate mammogram images. A Unet-based generator was used to produce synthetic images for multiple classes. A CNN model was employed to build a discriminator model in order to authenticate the synthetic images. Thus, extended training was not required for image generation.

### 2.4. Feature Engineering

The authors constructed a CNN model with four convolutional layers, batch normalization, and Leaky ReLu. Figure 3 highlights the proposed feature engineering process for extracting crucial features. The EfficientNet B7 model’s weights were used for the feature extraction. 

EfficientNet B7’s architecture relies heavily on inverted residuals. It uses a lightweight depth-wise separable convolution, succeeded by a linear bottleneck and a skip connection. This architecture facilitates the efficient capturing of intricate patterns. Compound scaling scales the model in depth, width, and resolution. It ensures that the model’s power increases with its size while retaining its efficiency. Mobile inverted convolution is a composite operation that integrates the principles of depth-wise separable convolution and linear bottleneck. This procedure possesses high computational efficiency and contributes to reducing the number of parameters while preserving meaningful patterns. 

The EfficientNet models were trained using the larger datasets. The authors froze the lower-level layers of the EfficientNet B7 model to prevent overfitting and promote better generalization. The QAT strategy was used to improve the efficiency of the proposed BC detection model [31]. This prepares the proposed model for deployment in resource-constrained environments. In addition, it reduces the computational cost and memory requirements of the proposed model. The authors applied dropout layers to address overfitting by using assigned probabilities. The extracted features were flattened and forwarded to fully connected (FCN) layers. The authors integrated the FCN layers to collect the outcomes for each class.

### 2.5. BC Classification

LightGBM is an ensemble learning-based gradient-boosting or tree-based learning model. It improves prediction efficiency, manages extensive datasets, and minimizes training time. It is commonly suggested for the analysis of tabular datasets. LightGBM uses leaf-wise splits to generate more complex trees, reducing loss while strengthening accuracy. The partitioning process uses a unique sampling technique called gradient-based one-side sampling (GOSS). This method involves excluding data points with tiny gradients and utilizing the rest of the data for estimating information gain and facilitating tree development. Equation (2) shows the mathematical form of computing the gradients of the loss function.
(2)$$\left|g_{bc}\right|=\left|g_{bc}\right|+\lambda \times max\left(\left|g_{bc}\right|,\gamma \right)\;$$
where $\left|g_{bc}\right|$ is the gradient of the loss function in classifying BC, λ is the variable to control the number of one-side sampling, and γ is the gradient threshold.

After the training phase, the FCN layer was replaced with LightGBM to achieve optimal accuracy. The authors utilized the GOSS technique to classify features. The proposed technique excludes smaller gradients and considers the remaining features to generate an outcome. It can produce significant information gain in resource-constrained environments. In addition, it requires less training time and low memory and is compatible with smaller and larger datasets. Figure 4 highlights multi-class classification using the LightGBM model. Equation (3) shows the LightGBM-based BC detection model.
(3)$$Classification=LightGBM\;\left(X_{i},\;L,\;N,\;M\right),\;i=1,\;2,\;\ldots \;n$$
where X is the BC image feature, L is the learning rate, n is the number of estimators, and M is the maximum depth.

To fine-tune the performance of the proposed BC model, the authors applied Optuna hyperparameter optimization. Optuna is an open-source framework that can be implemented in the TensorFlow and PyTorch frameworks. It employs tree-structured parzen estimators to determine the hyperparameters of the LightGBM model. Equation (4) outlines Optuna hyperparameter optimization.
(4)$$\mathrm{Best}\_\mathrm{Parameters}=\mathrm{Optuna}\left(\mathrm{LightGBM}\left(\mathrm{Parameters}\right)\right)$$
where Best_Parameters is the optimized hyperparameters of the LightGBM model, Optuna is the hyperparameter optimization function, and LightGBM is the BC classification model.

### 2.6. Performance Evaluation

The authors employed multiple evaluation metrics to measure the performance of the BC detection model. The accuracy (Acc_y_) metric identifies the model’s capability to detect BC. Table 2 describes the notations of true positives (TPs), true negatives (TNs), FPs, and FNs. Equation (5) presents the mathematical form of the accuracy metric.
(5)$$Accuracy=\frac{B_{TP}+B_{TN}}{B_{TP}+B_{TN}+B_{FP}+B_{FN}}$$

The precision (Pre_n_) metric determines the number of FPs in the outcome, whereas the recall (Rec_l_) metric measures the model’s ability to find TPs. F1-Score (F1) is used to ensure that there is no uneven class distribution in the outcome. Equations (6)–(8) outline the computational forms of Pre_n_, Rec_l_, and F1.
(6)$$Pre_{n}=\frac{B_{TP}}{B_{TP}+B_{FP}}$$
(7)$$Rec_{n}=\frac{B_{TP}}{B_{TP}+B_{FN}}$$
(8)$$F1=\frac{2\times Pre_{n}\times Rec_{n}}{\left(Pre_{n}+Rec_{n}\right)}$$

Furthermore, the authors used Cohen’s Kappa (K) and Matthew correlation coefficient (MCC) to evaluate the multi-class classification performance of the proposed model. The confidence interval (CI), SD, and computational loss were calculated to measure the uncertainties of the proposed model. Equations (9) and (10) present the mathematical forms of K and MCC.
(9)$$K=\frac{2\times \left(B_{TP}\times B_{TN}-B_{FN}\times B_{FP}\right)}{\left(B_{TP}+B_{FP}\right)+\left(B_{FP}+B_{TN}\right)+\left(B_{TP}+B_{FN}\right)+\left(B_{FN}+B_{TN}\right)}$$
(10)$$MCC=\frac{B_{TN}\times B_{FP}-B_{FN}\times B_{FP}\;}{\sqrt{\left(B_{TP}+B_{FP}\right)+\left(B_{TP}+B_{FN}\right)+\left(B_{TN}+B_{FP}\right)+\left(B_{TN}+B_{FN}\right)}}$$

## 3. Results

The authors employed Windows 10 Professional, Intel i7 with 16 GB RAM, and NVIDIA GeForce RTX 3050 to implement the proposed BC detection model. The source codes of the EfficientNet B7 and LightGBM models were extracted from the Github repositories [32,33]. The Optuna algorithm was used to fine-tune the hyperparameters of the LightGBM model. The PyTorch 2.0 and TensorFlow v2.15.0 libraries were used to implement the proposed model. The computational settings for the model implementation are presented in Table 3.

Table 4 outlines the findings of the proposed model’s performance in the analysis of the CBIS-DDSM dataset. The suggested QAT strategy supported the proposed feature extraction to extract meaningful features. The model produced an optimal outcome for each class. 

Table 5 reveals a significant improvement in the multi-class classification ability of the proposed model. The recommended image preprocessing and augmentation techniques assisted the EfficientNet B7 model in extracting key features. In addition, the suggested hyperparameter optimization technique improved the proposed model’s classification accuracy.

The outcomes of the model’s performance using the combined dataset are presented in Table 6. The QAT strategy assisted the recommended model in achieving a better outcome in the combined dataset. In addition, the proposed feature extraction improved the BC detection model’s efficiency. Figure 5 highlights the performance outcomes of the proposed model.

Table 7 highlights the batch-wise performance of the suggested BC detection model on the CBIS-DDSM dataset. The recommended EL approach improved the performance of the proposed model in addressing the existing limitations of BC detection. 

Table 8 reveals the batch-wise performance of the proposed BC detection model. The model achieved reasonable results in a batch size of 12. However, the authors extended the analysis to a batch size of 16. The QAT strategy helped the proposed model to produce better results by addressing data imbalance. 

The suggested model’s batch-wise performance is listed in Table 9. It achieved a significant improvement at batch 16. The fine-tuned LightGBM model classified the BC images with an optimal accuracy.

The generalization of the BC detection model to the CBIS-DDSM dataset is presented in Table 10. The proposed model outperformed the existing models, obtaining a promising result. The fine-tuned LightGBM model classified the images with optimal accuracy. 

Table 11 presents the outcome of generalizing the proposed BC model to the CMMD dataset. The number of images in the CMDD dataset is larger compared to the CBIS-DDSM dataset. However, the proposed model obtained a superior outcome compared to existing models. 

The outcomes of comparative analysis using the combined dataset are revealed in Table 12. The proposed BC detection model addresses the challenges in detecting BC using BC images using the fine-tuned EfficientNet B7 and LightGBM models. 

Table 13 highlights the computational complexities of BC detection models. The proposed BC model requires fewer parameters and floating-point operations (FLOPs) for image classification. Inception V3 and EfficientNet B7 needed additional parameters to achieve a reasonable outcome.

Lastly, Table 14 shows the statistical significance of BC detection models. The findings highlight the reliability and trustworthiness of the proposed model. Based on the outcomes, the proposed model can handle out-of-distribution samples and produce reliable results in a real-life environment. In addition, healthcare practitioners can make effective decisions using the proposed BC detection model. 

## 4. Discussions

The authors built a BC detection model to identify BC in resource-constrained environments. The proposed model includes image preprocessing and augmentation techniques, feature engineering, and classifier models. Using the image preprocessing technique, the authors enhanced image quality. An image augmentation technique was used to improve the prediction accuracy of the proposed model. CNN-based feature engineering was used to extract the key features of BC images. The authors used the weights of the EfficientNet B7 model for feature extraction. The Optuna algorithm was employed to fine-tune the hyperparameters of the LightGBM model. 

Batch-wise performance analysis outcomes suggest that the model effectively addresses overfitting and class imbalance. Table 10, Table 11 and Table 12 highlight the findings of generalizing the BC model to the CBIS-DDSM, CMMD, and Combined datasets. It is evident that the proposed model obtained a superior outcome. The suggested image preprocessing and feature extraction techniques significantly improved the proposed model’s classification performance. Table 13 reveals the computational complexities of implementing BC detection models. In all three datasets, the proposed model demanded few parameters and FLOPs to achieve an accuracy of 99.4%, 99.9%, and 97.0% and a K value of 96.9%, 96.9%, and 94.1%. In contrast, existing models required a higher number of parameters for feature extraction. The authors employed the Optuna algorithm to fine-tune the LightGBM model’s hyperparameters to classify BC images in a resource-constrained environment. Table 14 discusses the statistical significance of BC detection models. The proposed model generated outcomes with minimal losses of 0.4, 0.3, and 0.3, SD of 0.0003, 0.0002, and 0.0003, and CI range of [98.1–98.5], [97.6–98.2], and [98.1–98.4] using the three datasets. The minimal computational loss and optimal accuracy revealed that there is no overfitting in the proposed model.

The findings reveal that the model is reliable and can handle variations in BC images. The proposed model can seamlessly integrate into the clinical workflow and support healthcare professionals in making effective decisions. The suggested approach can reduce the risk of errors and improve the overall outcome of the proposed BC detection model. The fine-tuned LightGBM model reduced the number of FPs and FNs and accurately identified normal, malignant, and benign images. The BC detection model enables healthcare providers to prioritize patient care, consultation, and complex decision-making while effectively managing routine assessments. It achieves a high performance level and contributes to BC’s reliable and standardized identification. Its few requirements for retesting, greater efficiency, and the possibility of earlier intervention can lead to overall cost-effectiveness. To improve BC screening services, the suggested BC detection model can be a valuable tool to supplement existing expertise. By facilitating prompt treatment with the potential for improved outcomes, early and precise detection can empower patients.

The authors generalized the model using three datasets. The generalization findings highlighted the significance of the suggested BC detection model in detecting benign, malignant, and normal images. Feature extraction assisted the LightGBM model in producing a superior outcome. Integrating diverse images into the combined dataset reinforced the model’s ability to identify BC’s intricate patterns. The inherent features of QAT strategies contributed to a significant performance improvement. The suggested feature extraction technique enabled the LightGBM model to overcome class imbalances. The regularization features of the LightGBM model supported the proposed model in generalizing to unseen images. 

El Houby and Yassin [9] classified malignant and nonmalignant breast lesions using a CNN model. The model required a larger number of parameters and FLOPs to classify the images. Umer et al. [12] built a BC detection model using convoluted features. Falconi et al. [22] fine-tuned pretrained models for classifying BC images. Pretrained models are designed for a specific set of classes. They may face challenges in handling multiple classes of BC. Agarwal et al. [34] used the Inception V3 and ResNet 50 models for BC detection. The complex architecture of these CNN models required additional training time and a larger number of parameters for BC detection. In addition, the ResNet 50 model demands huge computational resources to store and process the parameters. Zahoor et al. [35] applied a deep neural network and entropy-controlled whale optimization algorithms to classify BC images. They used Whale optimization for feature extraction. The MobileNet V2 and NasNet models were used for classification. The integration of CNN models and the implementation of Whale optimization required additional memory resources. Boudouh et al. [36] employed a TL technique to detect abnormalities in BC images. Bai et al. [37] developed a BC detection model using a feature fusion Siamese network. Their model faced challenges in capturing minor details of BC. The limitations of CNN models reduced their performance in BC detection. 

Furthermore, Bobowicz et al. [38] employed an attention-based DL system for BC classification. Levy and Jain [39] used the CBIS-DDSM dataset to train the VGG-16, ResNet 50, and Inception V3 models. They achieved an average accuracy of 84.16%. Ahmed et al. [40] proposed semantic and instance segmentation techniques for classifying BC images. Hameed et al. [41] proposed a BC detection model using histopathology images. In addition, the authors employed recent models, including Yolo-V8, CatBoost, and ShuffleNet V2, for comparison analysis using combined datasets. The current models required additional computational resources, restricting them from producing an exceptional outcome. Compared to our proposed BC detection model, the architecture of Inception v3 is complex, which may affect the interpretation of the outcomes. Implementing a BC detection model based on Inception V3 may demand substantial computational resources. Extended training is required for the Inception V3 model to produce a reasonable outcome. The EfficientNet B7 model demands considerable computation time for image classification. Training and deploying the model may require significant computer power and memory, which can be resource-intensive. In order to maintain the configurations of EfficientNet B7, a considerable amount of memory is required. This can make its implementation less straightforward on devices with limited memory capacity. In contrast, the recommended BC detection model requires limited computational resources for BC detection. It achieved an extraordinary outcome by addressing the existing limitations using effective image preprocessing and augmentation techniques. 

To deploy the proposed BC detection model, healthcare centers require DL professionals. The proposed BC detection model may demand additional training to streamline the detection process in a real-life environment. In real-time data, mammography images may include unique sizes, noises, and artifacts. A substantial image preprocessing technique is needed to improve EfficientNet B7-based feature extraction. Fine-tuning processes are required to implement the recommended model in resource-constrained environments. In addition, healthcare centers should integrate an application into the proposed model to maintain patient data privacy and security.

The proposed model produced promising results in detecting BC. However, it has certain limitations. Massive volumes of labeled data may be essential in order to train the suggested model to generalize effectively to novel and unseen cases. An imbalance in class distribution can impact the model’s accuracy. The model’s efficacy may vary depending on the population, ethnicity, or imaging modalities. Continuous monitoring is required to generalize the proposed BC detection model to a broader population. It is necessary to update the proposed model to handle new information as healthcare data evolve over time. In order to promote its smooth adoption, the proposed model should align with healthcare practitioners’ specific requirements. In the future, the performance of the proposed BC model can be improved using a graph convolutional network. 

## 5. Conclusions

The proposed study addressed the existing limitations in detecting BC using mammography images. Early detection of BC can reduce the mortality rate and support physicians to provide effective treatments. The authors followed an image preprocessing technique to enhance image quality. Key features were generated using the EfficientNet B7 model. The fine-tuned LightGBM detected BC with limited computational resources. The authors generalized the proposed model using the CBIS-DDSM and CMMD datasets. The experimental outcome revealed a significant improvement in the performance of the proposed model. The proposed model obtained an exceptional result compared to other recent models. The findings highlighted that the model can be implemented in healthcare centers to identify BC in earlier stages. The authors faced a few challenges in deploying the proposed model. Substantial training is required to overcome the shortcomings of mammography images. The complexity of EfficientNet B7 may reduce the performance of the feature extraction process. Generalizing the proposed model using diverse datasets can minimize the limitations in identifying crucial patterns of BC. However, the authors improved the training process using the QAT approach. The proposed model can be extended using graph convolutional networks. 

## Figures and Tables

**Figure 1 diagnostics-14-00227-f001:**
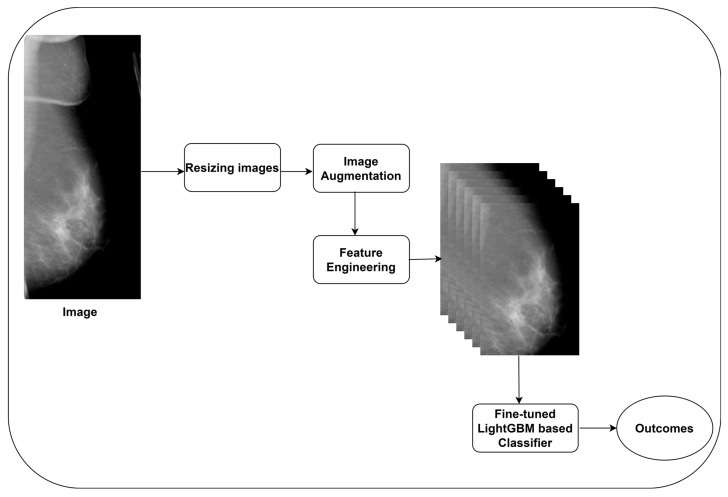
The proposed BC detection model.

**Figure 2 diagnostics-14-00227-f002:**
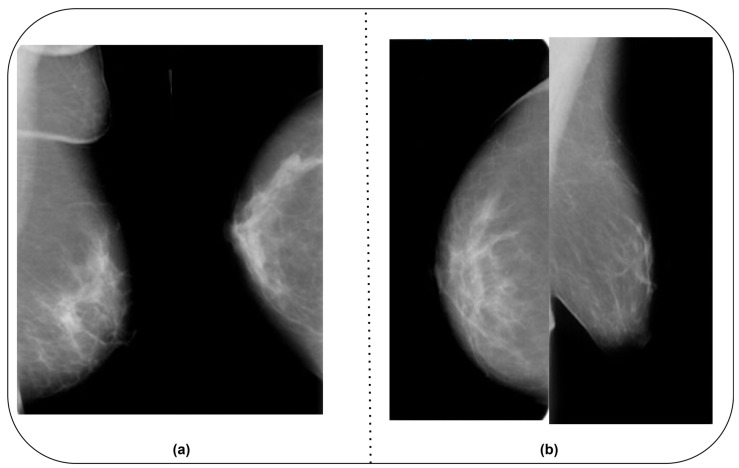
(**a**) CBIS-DDSM dataset. (**b**) CMMD dataset.

**Figure 3 diagnostics-14-00227-f003:**
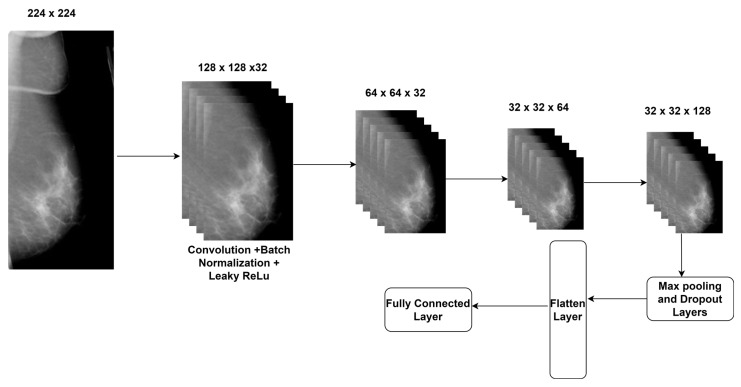
Feature engineering process.

**Figure 4 diagnostics-14-00227-f004:**
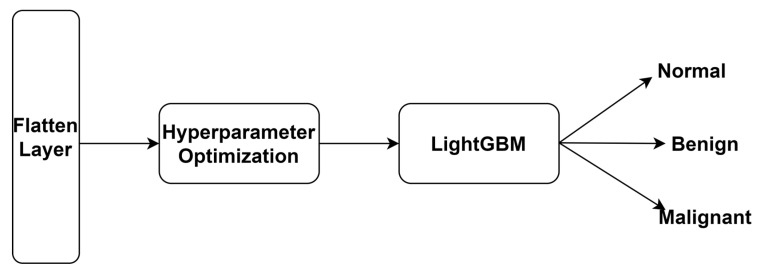
Fine-tuned LightGBM-based multi-class classification.

**Figure 5 diagnostics-14-00227-f005:**
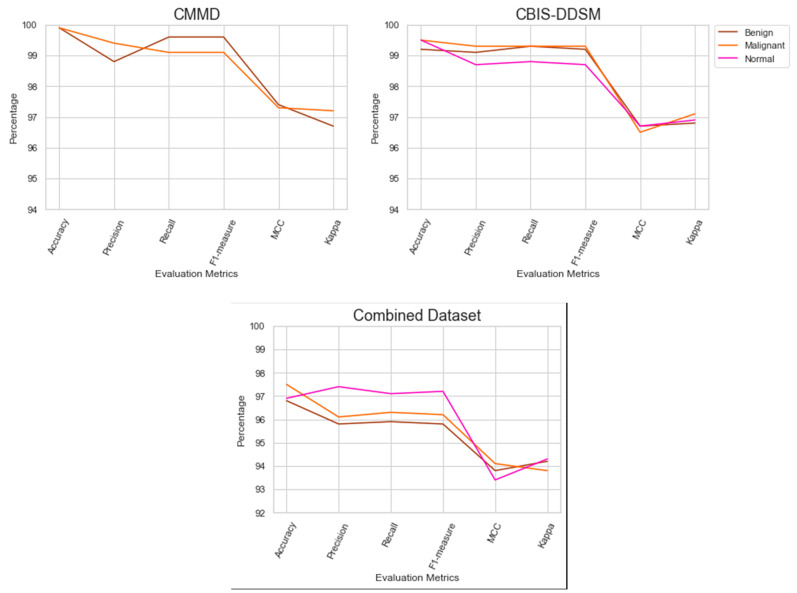
Findings of performance analysis.

**Table 1 diagnostics-14-00227-t001:** Dataset characteristics.

Datasets	Year	Number of Images	Number of Patients	Classification
CBIS-DDSM	2017	1644	2890	Benign, malignant, and normal
CMMD	2022	2214	1026	Benign and malignant
Combined Dataset	-	4900	-	Benign, malignant, and normal

**Table 2 diagnostics-14-00227-t002:** Notations and definitions.

Notations	Definition
$B_{TP}$	Correctly identified BC images
$B_{TN}$	Correctly identified normal or benign images
$B_{FP}$	Incorrectly identified BC images
$B_{FN}$	Incorrectly identified normal or benign images

**Table 3 diagnostics-14-00227-t003:** Computational settings.

Parameters	Values
BC image size	224 × 224
Strides	3
Number of convolutional layers	6
Learning rate	1 × 10^−4^
Decay rate	0.97 per 2 Epochs
Batch size	128
Fully connected layer	1
Activation function	Softmax

**Table 4 diagnostics-14-00227-t004:** Performance analysis outcomes—CBIS-DDSM.

Classes	Acc_y_	Pre_n_	Rec_l_	F1	K	MCC
Benign	99.2	99.1	99.3	99.2	96.8	96.7
Malignant	99.5	99.3	99.3	99.3	97.1	96.5
Normal	99.5	98.7	98.8	98.7	96.9	96.7
Average	99.4	99.0	99.1	99.1	96.9	96.6

**Table 5 diagnostics-14-00227-t005:** Performance analysis outcomes—CMMD.

Classes	Acc_y_	Pre_n_	Rec_l_	F1	K	MCC
Benign	99.9	98.8	99.6	99.6	96.7	97.4
Malignant	99.9	99.4	99.1	99.1	97.2	97.3
Average	99.9	99.1	99.3	99.2	96.9	97.3

**Table 6 diagnostics-14-00227-t006:** Performance analysis outcomes—combined dataset.

Classes	Acc_y_	Pre_n_	Rec_l_	F1	K	MCC
Benign	96.8	95.8	95.9	95.8	94.2	93.8
Malignant	97.5	96.1	96.3	96.2	93.8	94.1
Normal	96.9	97.4	97.1	97.2	94.3	93.4
Average	97.0	96.4	96.4	96.4	94.1	93.7

**Table 7 diagnostics-14-00227-t007:** Batch-wise performance analysis—CBIS-DDSM.

Batches	Acc_y_	Pre_n_	Rec_l_	F1	K	MCC
4	97.1	97.8	97.6	97.7	92.5	91.8
8	97.8	98.1	98.5	98.3	93.4	92.8
12	98.1	98.6	98.3	98.4	94.6	93.7
16	98.4	98.9	98.8	98.8	94.8	95.1
18	99.4	99.0	99.1	99.1	96.9	96.6

**Table 8 diagnostics-14-00227-t008:** Batch-wise performance analysis—CMMD.

Batches	Acc_y_	Pre_n_	Rec_l_	F1	K	MCC
4	97.5	97.5	97.2	97.3	93.4	94.1
8	97.9	96.9	98.2	97.5	94.7	95.1
12	98.8	98.6	98.6	98.6	95.3	96.4
16	99.9	99.1	99.3	99.2	96.9	97.3

**Table 9 diagnostics-14-00227-t009:** Batch-wise performance analysis—combined dataset.

Batches	Acc_y_	Pre_n_	Rec_l_	F1	K	MCC
4	95.2	94.2	94.5	94.3	93.1	93.4
8	95.8	94.9	94.7	94.8	92.8	92.1
12	96.4	95.6	95.3	95.4	93.5	92.6
16	97.0	96.4	96.4	96.4	94.1	93.7

**Table 10 diagnostics-14-00227-t010:** Findings of comparative analysis—CBIS-DDSM.

Models/Metrics	Acc_y_	Pre_n_	Rec_l_	F1	K	MCC
Proposed BC detection model	99.4	99.0	99.1	99.1	96.9	96.6
Falconi et al., model [22]	95.8	95.5	95.7	95.6	89.1	88.5
Agarwal et al., model [34]	96.7	96.9	96.9	96.9	91.2	89.4
Zahoor et al., model [35]	96.7	95.7	94.9	95.3	90.7	90.4
EfficientNet B7	97.1	96.8	97.3	97.0	92.3	91.7

**Table 11 diagnostics-14-00227-t011:** Findings of comparative analysis—CMMD.

Models/Metrics	Acc_y_	Pre_n_	Rec_l_	F1	K	MCC
Proposed BC detection model	99.9	99.1	99.3	99.2	96.9	97.3
Boudouh et al., model [36]	99.8	99.4	99.6	99.5	90.5	89.4
Bai et al., model [37]	98.4	98.2	98.2	98.2	90.8	88.7
Inception V3	96.7	97.5	96.8	97.1	91.1	90.3
Bobowicz et al., model [38]	97.1	97.2	97.1	97.1	91.3	90.6
EfficientNet B7	97.5	97.4	97.3	97.3	92.1	90.8

**Table 12 diagnostics-14-00227-t012:** Findings of comparative analysis—combined dataset.

Models/Metrics	Acc_y_	Pre_n_	Rec_l_	F1	K	MCC
Proposed BC detection model	97.0	96.4	96.4	96.4	94.1	93.7
Yolo—V8 model	92.4	90.5	90.7	90.6	89.2	90.4
El Houby and Yassin, model [9]	96.2	95.2	95.2	95.2	93.7	91.3
Singh et al., [13]	84.3	81.6	82.2	81.9	79.8	75.8
CatBoost model	95.1	94.9	94.6	94.7	92.8	89.9
ShuffleNet V2 model	91.3	90.8	91.2	91.0	90.2	88.5

**Table 13 diagnostics-14-00227-t013:** Computational strategies.

Models	CBIS-DDSM			CMMD			Combined Dataset		
	Learning Rate	Parameters (in Millions (m))	FLOPs (in Giga (G))	Learning Rate	Parameters (in Millions (m))	FLOPs (in Giga (G))	Learning Rate	Parameters (in Millions (m))	FLOPs (in Giga (G))
Proposed BC detection model	1 × 10^−4^	21	1.8	1 × 10^−4^	18	1.4	1 × 10^−4^	27	2.3
Falconi et al., model [22]	1 × 10^−2^	48	4.1	-	-	-	-	-	-
Agarwal et al., model [34]	1 × 10^−2^	34	3.4	-	-	-	-	-	-
Zahoor et al., model [35]	1 × 10^−4^	71	3.2	-	-	-	-	-	-
EfficientNet B7	1 × 10^−3^	31	2.2	1 × 10^−4^	27	1.9	-	-	-
Boudouh et al., model [36]	-	-	-	1 × 10^−3^	56	5.1	-	-	-
Bai et al., model [37]	-	-	-	1 × 10^−3^	65	3.8	-	-	-
Inception V3	1 × 10^−3^	75	3.7	1 × 10^−3^	58	4.1	-	-	-
Bobowicz et al., model [38]	-	-	-	1 × 10^−3^	69	5.2	-	-	-
Yolo—V8 model	-	-	-	-	-	-	1 × 10^−4^	38	4.2
El Houby and Yassin, model [9]	-	-	-	-	-	-	1 × 10^−3^	43	3.7
Singh et al., [13]	-	-	-	-	-	-	1 × 10^−3^	48	4.1
CatBoost Model	-	-	-	-	-	-	1 × 10^−4^	31	2.7
ShuffleNet V2 model	-	-	-	-	-	-	1 × 10^−4^	52	3.9

**Table 14 diagnostics-14-00227-t014:** Findings of uncertainty analysis.

Models	CBIS-DDSM			CMMD			Combined Dataset		
	Loss	CI	SD	Loss	CI	SD	Loss	CI	SD
Proposed BC detection model	0.4	[98.1–98.5]	0.0003	0.3	[97.6–98.2]	0.0002	0.3	[98.1–98.4]	0.0003
Falconi et al., model [22]	0.8	[96.1–96.7]	0.0004	-	-	-	-	-	-
Agarwal et al., model [34]	1.2	[95.8–96.2]	0.0005	-	-	-	-	-	-
Zahoor et al., model [35]	0.9	[95.9–96.4]	0.0005	-	-	-	-	-	-
EfficientNet B7	0.7	[97.1–97.6]	0.0003	0.8	[97.5–98.3]	0.0003	-	-	-
Boudouh et al., model [36]	-	-	-	1.4	[96.1–96.7]	0.0005	-	-	-
Bai et al., model [37]	-	-	-	1.1	[96.8–97.7]	0.0002	-	-	-
Inception V3	0.7	[97.3–98.2]	0.0003	0.7	[95.7–96.6]	0.0002	-	-	-
Bobowicz et al., model [38]	-	-	-	0.9	[97.1–97.8]	0.0004	-	-	-
Yolo—V8 model	-	-	-	-	-	-	0.9	[95.3–96.4]	0.0003
El Houby and Yassin, model [9]	-	-	-	-	-	-	1.2	[96.1–96.5]	0.0007
Singh et al., [13]	-	-	-	-	-	-	1.4	[95.8–96.3]	0.0005
CatBoost Model	-	-	-	-	-	-	0.5	[95.3–96.1]	0.0003
ShuffleNet V2 model	-	-	-	-	-	-	0.7	[95.4–96.2]	0.0004

## Data Availability

1. CBIS-DDSM Dataset. Available online: https://www.kaggle.com/datasets/awsaf49/cbis-ddsm-breast-cancer-image-dataset (accessed on 3 March 2023). 2. CMMD Dataset. Available online: https://wiki.cancerimagingarchive.net/pages/viewpage.action?pageId=70230508 (accessed on 5 March 2023).

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
