# Peer review of "An Enhanced LightGBM-Based Breast Cancer Detection Technique Using Mammography Images"

_diagnostics, 2024, doi:10.3390/diagnostics14020227_

Round 1

Reviewer 1 Report

Comments and Suggestions for Authors

Doctors examine the original images and decide for the patient. The doctor evaluates the raw X-ray image obtained.

Our main goal is to learn the doctor's knowledge and skills through artificial intelligence algorithms. Therefore, the proposed algorithms need to be trained on original images.

Pre-processing algorithms should not have been applied to the images.

It is not realistic for the accuracy value to be 99%. The accuracy may be high for the two datasets used, but this is inaccurate for the real environment. These results give rise to the statement "We don't need doctors."

The author should train by combining two data sets or use different test data. High training accuracy does not mean that the test accuracy of the system will be high. The author must determine the accuracy for different test images.

The Ensemble technique combines the results of multiple models and produces results. However, the author obtains the feature vector by using only a single model. Models in references 7,9 and 10 are ensemble models. However, the model proposed in the article is not ensemble-based.

There is no need for Figures 6 and 7.

Author Response

Doctors examine the original images and decide for the patient. The doctor evaluates the raw X-ray image obtained. Our main goal is to learn the doctor's knowledge and skills through artificial intelligence algorithms. Therefore, the proposed algorithms need to be trained on original images.

Response: As per the suggestion, we evaluated the proposed model using the combined dataset.

Pre-processing algorithms should not have been applied to the images.

Response: We thank you for the comments. Based on the comments, we removed Eqn. 2-4 from the preprocessing. However, we retained Eqn. 1 for resizing the contrast increased images. The raw mammography images require resizing and contrast enhancement to support the EfficientNet B7 based feature extracton. For instance, the EfficientNet B7 model requires an image size of 224 × 224. In addition, CLAHE improves the image quality.

It is not realistic for the accuracy value to be 99%. The accuracy may be high for the two datasets used, but this is inaccurate for the real environment. These results give rise to the statement "We don't need doctors." The author should train by combining two data sets or use different test data. High training accuracy does not mean that the test accuracy of the system will be high. The author must determine the accuracy for different test images.

Response: As per the suggestion, we combined the datasets and included mammography images of MIAS and BCDR-D datasets. We presented the details in line number 149. In addition, We employed GANs to generate synthetic images of benign, malignant, and normal image. One of the authors Dr. Ramprasad N who have expertise in medical imaging authenticated the images. In addition, he assisted me in evaluating and restructuring the article.

The Ensemble technique combines the results of multiple models and produces results. However, the author obtains the feature vector by using only a single model. Models in references 7,9 and 10 are ensemble models. However, the model proposed in the article is not ensemble-based.

Response: We thank you for the valuable suggestions. We removed the references and included references related to the proposed model.

There is no need for Figures 6 and 7.

Response: We thank you for your suggestion. We have removed the figures 6 and 7.

Reviewer 2 Report

Comments and Suggestions for Authors

It's an interesting article. I have several questions/suggestions

- The introduction section has too many references, and some of them should be in the discussion section. Please reorganise.

- I found the generalizability of results is lacking in the discussion section, please add.

- The author should compare more extensively the proposed model with existing techniques in terms of accuracy, efficiency, and practical deployment.

mute

Author Response

It's an interesting article. I have several questions/suggestions

The introduction section has too many references, and some of them should be in the discussion section. Please reorganise.

Response: We thank you for your suggestion. We have reorganized the introduction part and discussed the existing models in line number 371.

I found the generalizability of results is lacking in the discussion section, please add.

Response: In order to generalize the proposed model using diverse images, we have included the combined dataset. The generalized findings are presented in Tables 9 and 12-14. In addition, the generalization findings are discussed from line number 362.

The author should compare more extensively the proposed model with existing techniques in terms of accuracy, efficiency, and practical deployment.

Response: We thank you for your suggestion. We have compared the proposed model with recent models (Yolo-V8, Catboost, and existing breast cancer detection models ) and findings are presented in Table 12. In addition, the efficiency and practical deployment is discussed from line number 409.

Round 2

Reviewer 1 Report

Comments and Suggestions for Authors

Thank you for implementing the suggestions into your article.

Author Response

Dear Editor and Reviewers,

I thank you for your valuable suggestions for improving the article standards. I addressed the reviewers' comments. I highlighted the specific changes and responses in the following part of this document.

Academic Editor:

  1. The title should be revised to reflect mammography.

Response: We thank you for the suggestion. We have modified the title to reflect mammography images.

  1. The title is suggested to be revised to reflect the specific deep learning technique.

Response: We thank you for the suggestion. We have modified the title that presents the specific deep learning technique.

  1. "detect BC using CNN and DL models": Is not CNN a DL technique? Please carefully proofread the manuscript.

Response: We thank you for the comments. We have carefully proofread the manuscript and addressed the mistakes.

  1. Why EfficientNet B7 is useful for the task involved?

Response: Compared to the traditional CNN model, EfficientNet B7 offers an impressive performance with few parameters. It can capture the intricate patterns from the mammography images. The scalable architecture of the EfficientNet B7 can assist the proposed model to generalize in real-time application. We have presented these sentences in line numbers 132 and 141.

  1. Values of accuracy like "99.4, 99.9" should be " 99.4%, 99.9%"

Response: We thank you for the comments. We have presented the outcomes with % in the necessary places.

  1. Please clarify the novelty of this manuscript. The contributions presented at the end of the Introduction section does not have much novelty.

Response: Thank you for the suggestions. We have clarified the study’s novelty in line numbers 103 and 109.

  1. Some of the figures are of poor quality. The resolution should be improved.

Response: As per the suggestion. We have improved the quality of the images.

  1. How to ensure there is no overfitting for the authors' model?

Response: The minimal computational loss and the optimal accuracy indicate that there is no overfitting in the model. In addition, the batch wise analysis outcome highlighted the absence of overfitting. The recommended data augmentation and QAT strategy has supported the model to overcome overfitting issues.

Reviewer 2:

Thank you for implementing the suggestions into your article.

Response: We thank you for your valuable suggestions.